# Group antenatal care positively transforms the care experience: Results of an effectiveness trial in Malawi

Crystal L. Patil[1]*, Kathleen F. Norr[2], Esnath Kapito[3], Li C. Liu[2], Xiaohan Mei[2], Elizabeth Chodzaza[3], Genesis Chorwe-Sungani[3], Ursula Kafulafula[3], Elizabeth T. Abrams[1], Allissa Desloge[4], Ashley Gresh[5], Rohan D. Jeremiah[2], Dhruvi R. Patel[6], Anne Batchelder[5], Heidy Wang[2], Jocelyn Faydenko[2], Sharon S. Rising[7], Ellen Chirwa[3]

1 University of Michigan, Ann Arbor, Michigan, United States of America, 2 University of Illinois Chicago, Chicago, Illinois, United States of America, 3 Kamuzu University of Health Sciences, Blantyre, Malawi, 4 University of North Carolina Charlotte, Charlotte, North Carolina, United States of America, 5 Johns Hopkins University, Baltimore, Maryland, United States of America, 6 Old Dominion University, Eastern Virginia Medical School, Norfolk, Virginia, United States of America, 7 Group Care Global, Philadelphia, Pennsylvania, United States of America

* clpatil@umich.edu

## Abstract

### Background

We developed and tested a Centering-based group antenatal (ANC) model in Malawi, integrating health promotion for HIV prevention and mental health. We present effectiveness data and examine congruence with only the Group ANC theory of change model, which identifies key processes as supportive relationships, empowered partners in learning and care, and meaningful services, leading to better ANC experiences and outcomes.

### Methods

We conducted a hybrid effectiveness-implementation trial at seven clinics in Blantyre District, Malawi, comparing outcomes for 1887 pregnant women randomly assigned to Group ANC or Individual ANC. Group effects on outcomes were summarized and evaluated using t-tests, Mann-Whitney, or Chi-squared tests. Adjusted for seven individual and two clinic-level baseline covariates, point estimates are reported for continuous outcomes using multivariable linear regression models. Adjusted for the same covariates, odds ratios are reported for categorical outcomes using logistic regression models. All statistical tests were two-sided, controlling for a Type I error probability of 0.01 due to multiple testing.

**Data availability statement:** Data are accessible in a public repository, University of Michigan - Deep Blue Data, at https://doi.org/10.7302/a315-2417

**Funding:** This study was funded via Grant #R01NR018115, awarded to the University of Illinois Chicago from the National Institutes of Health, National Institute for Nursing Research (NINR). The funder had no role in study design, data collection and analysis, decision to publish, or preparation of the manuscript.

**Competing interests:** The authors have declared that no competing interests exist.

### Findings

Women in Group ANC had higher peer connectedness and pregnancy-related empowerment, recalled receiving more services, and discussed more health promotion topics. They experienced less wait time, greater satisfaction with care (Estimate = 1.21, 99% CI = 0.07, 2.35), and had a higher mean number of ANC contacts (Estimate = 0.74, 99% CI = 0.50, 0.98). Most women preferred Group ANC for a future pregnancy (81% in Individual ANC; 95% in Group ANC). Women in Group ANC had more diverse diets (Estimate = 0.35, 99% CI = 0.17, 0.53), were better prepared for birth (Estimate 0.32, 99% CI = 0.16, 0.48), more likely to use condoms consistently (OR= 1.07, 99% CI = 1.00, 1.14) and communicated more with partners. They reported less mental distress in late pregnancy (Estimate = −0.61, 99% CI = −1.20, −0.02). Exclusive breastfeeding, partner HIV testing and disclosure, facility-based delivery, postnatal care attendance, postpartum family planning, and low birth weight did not differ by ANC type.

### Conclusions

This effectiveness study of Malawi Group ANC, the first trial with individual randomization in a low-income country, maps outcomes to the theory of change, enhancing our understanding of Group ANC's diverse positive impacts. The integration of typically neglected health promotion topics into the model highlights its flexibility to address changing local and global needs. Based on study results, the Malawi Ministry of Health is introducing Group ANC at district-level trainings and exploring the logistics for nationwide adoption. With momentum and political will, better care and a positive healthcare experience can be achieved for women, infants, and families in Malawi and globally.

### Trial registration

ClinicalTrials.gov NCT03673709.

### Introduction

Antenatal care (ANC) saves lives through screening and diagnosis, implementing preventive measures, and promoting health to support informed decision-making [1–4]. ANC is highly valued, however, in low- and middle-income countries (LMICs), only about half of pregnant women receive adequate care [5]. ANC is consistently described as care that feels rushed and impersonal with long waits for services, and health promotion is typically through a general didactic health lecture with minimal opportunity to address questions [6–10]. Even though pregnant women highly value ANC, their experiences do not reinforce this. ANC is most effective when the quality is high, and women return for care as expected [1].

Group ANC can address many of the quality-of-care issues in LMICs, including prolonged wait times, suboptimal patient-provider interactions, and missed essential

services and health promotion, all of which contribute to increased health risks and lower ANC attendance [6–10]. Centering-based Group ANC, developed by a US-based midwife [11,12], transforms care delivery by providing an interactive and efficient context for respectful care, thereby supporting a positive pregnancy experience [11–13]. In Group ANC, after the standard one-on-one initial visit, 8–12 women at similar gestational ages attend ANC together, guided by the same trained ANC provider and co-facilitator. The number and timing of contacts and required health promotion topics are the same as usual (individual) ANC. Excluding any wait time, Group ANC contacts last 1.5–2 hours, compared to 10–20 minutes for individual ANC contacts. There are three interrelated and theoretically informed [14–16] core components of the Centering-based group care model that act synergistically to enhance the ANC experience [11]. The first component is that recommended health assessments are completed with a provider in the group meeting space, and that the pregnant group members perform self-assessments (e.g., measure and record their own blood pressure and weight). The second component, interactive learning, supports health promotion and health-promoting behaviors by encouraging participation and respecting group members' contributions. Partners and non-pregnant members can also accompany the pregnant client. The continuity of care and increased time allow women to deeply engage in their care as they draw on and apply the collective knowledge and experiences of facilitators and group members to change behavior [11]. Third, community building among group members, co-facilitators, and beyond the healthcare moment is foundational to this model and is supported by continuity of group members and socializing across contacts to form social connections, relationships, and trust. A robust body of research documents positive outcomes in high-income settings, including reduced risk for prematurity and low birth weight [17–25].

Recommendations in the 2016 WHO ANC guidelines included an increase in the number of contacts from four to eight and an emphasis on respectful, person-centered care to support a positive pregnancy experience [1,26,27]. Based on the limited evidence available before 2016 [28–30], WHO recognized Group ANC as a promising health system intervention that could improve the utilization and quality of ANC but highlighted the need for rigorous research [1]. Since then, a growing body of research shows that Group ANC is feasible and acceptable in LMICs [31–37] and associated with high satisfaction with care and connections with peers and facilitators [13,38–51]. Group ANC is linked to increased ANC attendance [31,50–58] and other positive outcomes including health literacy [50,59], birth preparedness [50], partner communication [46], facility-based delivery [54], uptake of vaccinations [60], use of insecticide-treated nets [60], family planning uptake [59,61,62], and breastfeeding practices [63].

Despite the rapid expansion of research on Group ANC in LMICs, there has been less attention to identifying causal pathways through which group care substantially improves the wide diversity of reported outcomes. A comprehensive theory of change model incorporating the components of Centering-based group healthcare proposes multiple pathways that transform the care experience and link to outcomes [13]. Based on key themes identified in qualitative interviews with pregnant participants and their providers, this model captures how the Group ANC experience fosters three mutually reinforcing elements: developing supportive relationships, becoming empowered partners in learning and care, and meaningful clinical services and information. Together, these factors transform the ANC experience and produce a self-reinforcing cycle of more and better care leading to health knowledge, confidence, and healthy behaviors. This theory of change model makes an important contribution to the field because it captures the transformative nature of the group care process and is rooted in the experiences of group care clients and providers [13].

The Malawi Group ANC model reported here retains the core components of the evidence-based CenteringPregnancy model and includes two locally-relevant modifications [11,29]. The first modification was the type of non-provider co-facilitator. With encouragement from the Reproductive Health Unit and the Community Volunteers program in Malawi, trained community volunteers were asked to serve as co-facilitators with a midwife. Community volunteers have been widely used for government community programs but have not previously been incorporated into ANC in Malawi. The second modification was the addition of Malawi-specific health promotion topics of HIV prevention and mental health. In LMICs, mental health is not typically monitored as part of ANC. In addition to the stigmatization of mental health conditions

and lack of training, few providers or community health workers have the knowledge and experience to discuss mental health confidently [64–68]. Even though Malawi has a strong prevention of mother-to-child transmission program and HIV testing and counselling are a part of care, deep discussion of HIV prevention is not typically integrated into didactic ANC health lectures. Because women and girls account for 63% of all new infections in sub-Saharan Africa [69], we integrated an evidence-based HIV prevention component into the health promotion content [29]. Given Malawi's high rates of common mental disorders [70] and prevalence of HIV [69], and their frequent co-occurrence [71–73], we integrated these two topics into the Group ANC health promotion content.

The purpose of this manuscript is to present the clinical trial results assessing if Group ANC improves outcomes for pregnant women in Malawi and if the patterning is congruent with the theory of change model [13].

## Methods

We conducted a hybrid effectiveness-implementation trial [74] with two aims. Details summarizing the study's conceptual basis, methods, and design are in a protocol paper [75]. This manuscript reports the results of the first aim with data collected between 5 July 2019 and 10 August 2023, comparing outcomes for pregnant women randomly assigned to Group ANC or Individual ANC at seven clinics in Blantyre District, Malawi. The planned sample size was 1776 in the proposal based on the clustered study design for the detection of a reduced preterm birth rate from 18% in the individual care arm to 12% in the Group ANC arm, controlling for 80% statistical power and 30% retention.

### Site and setting

Malawi is a sub-Saharan African country with a free national healthcare system. Malawi has committed to Universal Health Coverage but spends less than 40 USD per capita on health, which is below average for the region and half of what is recommended by WHO for implementing a minimum set of essential interventions [76–78]. Less than 45% of facilities can comprehensively deliver the minimum package of free health services [76]. In addition to limited financing, Malawi has a severe health workforce shortage, with only 1.48 health workers per 1,000 people and a severe shortage of midwives, with only 40% of needed positions currently filled [79]. Malawi is among the countries hardest hit by the HIV epidemic and has a heavy burden of both communicable and non-communicable diseases [80,81]. Global programs have greatly improved HIV testing and treatment rates and health outcomes. In 2022, HIV prevalence in Malawi was estimated at 7.1% overall and 8.0% for adults. Rates, however, are not uniform: women and girls accounted for more than 60% of all new infections, and, at 17%, Blantyre District, had the highest HIV prevalence rate in the nation [82].

The seven study clinics were selected in consultation with the Blantyre District Health Team to represent a diversity of clinics and communities served. The catchment areas and populations served by the clinics differ socioeconomically. Three clinics serve the urban population of Blantyre city, the rapidly-growing second largest city in Malawi [83]. Two clinics serve the peri-urban communities adjacent to metropolitan Blantyre, and two clinics primarily serve a rural and predominately agricultural community. The clinics vary in volume and number of working midwives.

### Ethical approval

The Kamuzu University of Health Sciences (COMREC: P.10/18/2498) and the University of Illinois Chicago (IRB: 2018–0845) granted ethical approval for this study. Data were collected from women aged 15 and older and less than 24 weeks gestational age presenting for antenatal care at the seven study health centers. Those aged 15–17 assented with written consent from a legal guardian and those older than 17 provided written informed consent between 5 July 2019 and 10 August 2023.

### Participant recruitment, enrollment, and randomization

All pregnant women presenting for their first antenatal visit received the same standard individual intake visit that included a health assessment with the midwife, laboratory tests, and HIV testing. After completing the intake visit, midwives

directed clients to study team members so that eligibility could be assessed. Those under the age of 15, more than 24 weeks pregnant, or unable to make an informed choice (e.g., unable to converse about the study) were not eligible. Consented women then completed the baseline self-report survey using Audio Computer-Assisted Self-Interview software [84–87]. The study statistician determined the randomization order list for each site before recruitment, and assignments were placed in order in sealed envelopes. After completing the baseline survey, the woman selected the next sealed envelope in that clinic's box to reveal the type of ANC assignment, Group ANC or Individual ANC. Research assistants engaged in survey data collection were blinded to participant assignment.

## Study interventions

Both Individual ANC (control) and Group ANC (intervention) adhered to the Ministry of Health guidelines. Regardless of random assignment, the health assessments and services provided to clients were expected to follow these guidelines. The two study arms are described in detail below.

Individual ANC (control arm): Pregnant women randomly assigned to Individual ANC received the same ANC and postnatal care services typically provided to other clients. ANC services were provided on a first-come, first-served basis. Health lectures were presented by a midwife to those in the waiting area. Participants met individually with a midwife for health assessments at each contact. However, there was one important difference between study participants and pregnant women receiving usual ANC at each clinic. The Malawi Ministry of Health was planning to transition to the 2016 WHO recommendation of an 8-contact ANC model (vs 4-visit model), so they advised that both study conditions should be offered the opportunity to attend ANC eight times, especially since unequal numbers of contacts would weaken the evaluation of the impact of the type of ANC. Importantly, pregnant women not in our study followed the 4-visit focused antenatal care model and were not given specific recommended dates to return. Participants in the Individual ANC arm were encouraged to attend 8 contacts and provided a written schedule with the seven remaining dates they should attend. Therefore, the Individual ANC arm took on some characteristics of an intervention. As the project progressed, the 8 contact model was gradually rolled out in Malawi.

Group ANC (intervention arm): The Malawi Group ANC model for this study was adapted from the Centering-based Group ANC model [11,29,31]. After the initial individual intake visit, there were seven 2-hour antenatal contacts and one postnatal contact at approximately six weeks after birth, conducted on-site in the same clinics scheduled in advance with dates and times. Each contact followed the Centering-based pattern [11] with the first 30–45 minutes devoted to socializing and health assessments, including self-care activities (e.g., taking one's own blood pressure and weight) and a one-on-one health assessment with the midwife in the group space. As part of the one-on-one health assessment with the midwife, women could have confidential discussions to address matters of personal concern. Health issues requiring treatment or referral were managed outside the group, and emergencies like hypertension or malaria would be addressed immediately. Socializing primarily occurred as health assessments were conducted and continued throughout the session as the group gathered in a circle for 75–90 minutes of interactive health promotion activities. The recommended health promotion topics, aligned with gestational age, were discussed interactively and included games, role plays, and sharing of related knowledge and experiences. Health promotion topics were flexible and could be adjusted to meet group members' needs.

## Implementation strategies

Before launching, we updated the Implementation Toolkit used in the pilot study [29,31] to include a revised Facilitator guide, measures of fidelity, and the curriculum for the basic and mentor training workshops [75,88]. We used four strategies to support implementation. The first strategy was use of a 3-Step Implementation Model (prepare, rollout, and sustain) that demonstrated effectiveness in the successful scale-up of Kangaroo Mother Care in South Africa [89,90] and a community peer group HIV prevention program in Malawi [91]. This approach guided the overall implementation at each clinic. The second strategy was a robust training and mentoring program that emphasized fidelity and strategies to ensure

respectful care [88]. Initial concerns about discussing sensitive topics or health assessments in group space often diminish as trust builds and facilitators become more experienced [11,38,92–94]. Our training drew on successful training that is sensitive to and accommodates individual needs while promoting fidelity to the core components [11,12,88,95,96]. The toolkit included tracking forms for clinics and facilitators to self-assess fidelity, and track attendance and benchmarks. Training and mentoring ensured the development of in-country mentors who could train and support new Group ANC facilitators for sustaining and scaling up. The third strategy was to provide support and interactive assistance from the Implementation Coordinator to each clinic's implementation team; the coordinator facilitated problem-solving of implementation issues and adjustments of Group ANC to each clinic's specific context while being attentive to maintaining fidelity. We staggered rollout across 3 years to ensure the implementation team could provide ongoing support in person and by phone. Fourth, we minimized the need for additional resources and materials. To support future sustainability, the costs of training were absorbed by the project. Malawi now has a large cadre of trained midwives who can offer in-country training for Group ANC. Throughout the study, no major changes were made to the Group ANC intervention model or surveys. The only change in standard of care was the slow rollout of the transition from four recommended visits to eight contacts. However, this did not affect our study because we anticipated this change and adjusted the control arm so that women in both study arms had the opportunity for the same number of visits that they could attend. Basic obstetric ultrasound scanning services were also introduced at clinics, but individuals in both arms would have benefited equally from this quality improvement activity.

## Measures

Seven individual-level demographic and socioeconomic variables were treated as covariates in all analyses. Two clinic-level variables were also included as covariates, catchment area (rural, peri-urban, or urban) and midwife ANC workloads. Midwife ANC workloads were captured by a ratio of the average number of new ANC clients served each month divided by the number of midwives, with a higher ratio indicating a higher workload. The ratio was highest in two rural-serving clinics and substantially lower in peri-urban and urban communities. Table 1 defines the individual-level and clinic-level covariates and outcome variables and their linkages to the theory of change [13]. We originally intended to assess the impact of Group ANC on preterm birth rates, which requires an accurate assessment of gestational age at birth. After integrating all available data from health records and women's reports (i.e., last menstrual period, midwife assessments, ultrasound), we were still unable to obtain a reliable measure of gestational age at birth, even when ultrasound results were available. Therefore, we opted to use low birth weight as a proxy for this outcome.

## Procedure

Each participant was assigned an identification code, and the study team created a tracking system to contact participants and track progress through the trial. Data were obtained from four ACASI surveys completed by participants at baseline, late pregnancy, 6–8 weeks postpartum, and six months postpartum. To reduce bias, participants were not compensated for ANC visits; they were offered the Malawi Kwacha equivalent of five US dollars for the time and effort it took to complete the survey. To supplement survey responses about specific health outcomes, study staff extracted missing data from standard Ministry of Health antenatal, birth, neonatal, and postnatal registers; individual longitudinal antenatal and postnatal records; and through personal follow-up. Data cleaning checks (range and logic) were applied, and eligibility and critical outcomes data requiring further cleaning were completed by designated study staff to review and resolve discrepancies.

## Statistical analysis

Randomization and missing data were examined using t-tests, Mann-Whitney, or Chi-squared tests, according to the variable types and their distributions, to assess the balance between the two study arms for all baseline variables and the comparability between individuals with or without missing data. Descriptive analysis was performed for individual- and

**Table 1. Definitions of the individual-level and clinic-level covariates and the continuous and categorical outcome measures organized to align with the theory of change model [13].**

| VARIABLE | OPERATIONAL MEASURE |
|---|---|
| **Individual-level covariates** | |
| Age | A 3-category ordinal variable: (1) 15–19 years (adolescents); (2) 20–29 years; and (3) 30 years or more. Age 15–19 years is the reference category |
| Parity | A 2-category indicator is used: (1) Primiparity, indicating a first pregnancy, or (2) Multiparity, indicating one or more previous pregnancies. Primiparity is the reference category |
| In a relationship | Self-defined relationship status: (1) Not in a relationship or (2) In a relationship (married, living with a partner, or dating someone). Not in a relationship is the reference category |
| Education | A three-category ordinal variable: (1) Less than a primary education; (2) completed primary education; and (3) more than a primary education. Less than primary is the reference category |
| Food insecurity | Participants were asked whether their household had ever run out of food or money to buy food in the last 12 months. For this dichotomous variable, no food insecurity is the reference category |
| Ownership index | Participants were asked yes/no questions about having electricity or a generator and owning farm animals and agricultural land. A total of all the yes responses produced the index as an ordinal variable (0–3), with 0 serving as the reference category |
| Income quartile | Participants were asked to approximate the weekly income generated by the household. These values were converted to USD equivalents, corrected for the year the survey was completed. Quartiles were treated as an ordinal variable: (1) $0-$0.98; (2) $0.98-$2.88; (3) $2.88-$5.77; and (4)>=$5.77. Quartile 1 is the reference category |
| **Clinic-level covariates** | |
| Client-to-midwife ratio | Average number of new ANC clients per month divided by the number of midwives providing ANC. A higher ratio indicates a higher workload per midwife, range 7.4–29.0 |
| Community catchment area | A 3-category ordinal indicator of the community clinics served: (1) Rural (n = 2) have low population density and mainly agricultural economic activities; (2) Peri-urban (n = 2) are near urban centers with a mix of rural and urban characteristics; and (3) Urban (n = 3) are densely populated, with a more diverse economy. Rural is the reference category |
| ***Outcomes*** | |
| ***Supportive relationships, empowered partners in learning and care, meaningful services and information*** | |
| Peer connectedness | An index indicating the number of yes responses for 14 items about feelings and exchanges from relationships with other pregnant women at ANC (e.g., acceptance, belonging, and advice); range 0–14 |
| Pregnancy-related empowerment (PRES) | The 16-item Likert-type scale [97] with 1 (strongly disagree) to 4 (strongly agree) capturing a sense of control over pregnancy health and healthcare across four domains: provider connectedness, skillful decision-making, peer connectedness, and gaining voice; range 16–64, α = 0.93 |
| ANC services | A checklist indicating the number of recalled ANC services: weight, blood pressure, gestational age, fetal heart tones, fetal position at every visit, and receiving tetanus vaccine, iron tablets, and prophylaxis for malaria and worms; range 0–9 |
| ANC health promotion topics | A checklist indicating discussing 13 topics: healthy eating, danger signs, complications, malaria prevention, place of delivery, facility-based delivery, breastfeeding, early initiation of breastfeeding, birth spacing, family planning, safer sex, birth preparedness, and exclusive breastfeeding; range 0–13 |
| Repeat HIV test at ANC | If seronegative at intake, had at least one additional HIV test at ANC (yes/no). No is the reference category |
| ***ANC experience*** | |
| Wait time for ANC services | Wait time of more than one hour for ANC services (yes/no). Yes, more than one hour is the reference category |
| Satisfaction with ANC | A 10-item scale rating satisfaction with the overall quality of the care, the procedures, providers' level of respect and listening skills; 5-point Likert scale; range 10–50, α = 0.90 |
| ANC contacts | |
| Number of contacts | Total number of ANC contacts, 1–8+, mean and standard deviation |
| ANC4+ | Whether 4 or more ANC contacts were reported during this pregnancy (yes/no). No is the reference category |
| ANC8+ | Whether 8 or more ANC contacts were reported during this pregnancy (yes/no). No is the reference category |
| Prefer Group ANC in a future pregnancy | In late pregnancy, participants were asked if they would prefer Individual or Group ANC if they got pregnant in the future. Individual ANC is the reference category |

*(Continued)*

**Table 1.** (Continued)

| VARIABLE | OPERATIONAL MEASURE |
|---|---|
| **Knowledge** | |
| Healthy pregnancy knowledge | 25 yes/no items based on content supporting a healthy pregnancy, birth, and infant, such as danger signs, healthy eating, newborn care, birth spacing, family planning, and infant feeding; range 0–25 |
| UNAIDS knowledge, score | UNAIDS comprehensive HIV prevention knowledge [98] is the number of correct responses to five questions: if a healthy-looking person can have HIV, whether condom use and having only one uninfected partner reduce the likelihood of getting HIV transmission and two of the most common local misconceptions about HIV transmission (casual contact and mosquito bites) |
| UNAIDS knowledge, all correct | A yes/no indicator that all five UNAIDS questions were answered correctly. No, not all correct is the reference category |
| **Health-promoting behavior** | |
| Dietary diversity | 6 yes/no items indicated food categories eaten in the previous 24 hours: (1) ≥ 3 fruits or vegetables; (2) chicken, fish, eggs, insects or other meat; (3) groundnuts or groundnut powder, dried beans or peas, or soya; (4) leafy green vegetables; (5) yellow or orange fruits or vegetables; and (6) milk (fresh or fermented) or yogurt, range 0–6 |
| Birth preparedness | A checklist of 5 recommended behaviors, including if the woman talked to the midwife about where she planned to deliver, if she obtained supplies, arranged transportation, set aside money, and identified a blood donor, range 0–5 |
| Exclusive breastfeeding | Exclusive breastfeeding for six months (yes/no). No is the reference category |
| Consistent condom use | Participants reported whether a condom was used with every sexual encounter in the last two months (yes/no). No is the reference category |
| **Communication and care beyond ANC** | |
| Partner communication | 6-item yes/no index capturing whether the couple communicated about specific topics: (1) healthy diets; (2) exclusive breastfeeding; (3) difficult subjects; (4) family planning; (5) HIV testing; or (6) use of condoms; range 0–6, α = 0.99 |
| Partner HIV test disclosure | Whether a partner who was not HIV positive at the intake visit took an HIV test and shared results during this pregnancy (yes/no). No is the reference category |
| Facility-based delivery | Woman gave birth in a health facility (yes/no). No is the reference category |
| Postnatal care attendance | An indicator of attending ≥1 postnatal visit after discharge (yes/no). No is the reference category |
| Family planning | Reported use of family planning at late postpartum, approximately 6 months after delivery (yes/no). No is the reference category |
| **Clinical** | |
| Birth weight | Whether a newborn weighed <2,500 grams (5.5 pounds) at birth (yes/no) is used as a proxy for preterm birth due to the lack of accurate gestational age data. No is the reference category |
| Mental health | Self-Reporting Questionnaire has 20 yes/no items that screen for symptoms of anxiety, depression, psychosomatic symptoms, reduced vital energy, and depressive thoughts [70,99], range 0–20 with a higher score indicating higher mental distress, baseline α = 0.79; late pregnancy α = 0.85 |

clinic-level characteristics. Group effects on post-intervention outcomes were first summarized and evaluated using t-tests, Mann-Whitney, or Chi-squared tests. We next used multivariable regression models for continuous outcomes and logistic regression models for categorical outcomes to assess the effects of Group ANC. Since the participants were receiving care from different clinics, resulting in potential clustering of individuals, we first used mixed-effect regression models (MRM) or mixed-effects logistic regression models (MLRM) with random clinic (cluster) effects to assess within cluster correlations. After random clinic effects were deemed insignificant using likelihood ratio tests, we employed multivariable linear regression or logistic regression models for post-intervention outcomes to evaluate the intervention effectiveness, adjusted for the baseline outcome level and significant individual or clinic characteristics. In the results from linear regression models, point estimates of Group ANC versus Individual ANC differences and their 99% confidence intervals (CIs) were reported, while odds ratios (ORs) for the Group ANC effects, compared to the individual ANC, and their 99% CIs were reported. Model selections and key model assumptions examinations were performed for all statistical

models. Baseline individual and clinic covariates (Table 1) were included in all models. For pregnancy-related clinical outcomes analyses, additional covariates included typical risk factors such as women's height, HIV status, tobacco use, anemia (hemoglobin < 11g/dl), and dietary diversity. In addition to the typical covariates, peer connectedness was included as a covariate for the ANC quality indicators of satisfaction with ANC and type of ANC preference. All statistical tests were two-sided, controlling for a Type I error probability of 0.01 due to multiple testing of numerous outcomes. All data analyses were conducted using SAS 9.4 [100].

## Results

### Sample and randomization

A total of 8257 pregnant women were approached to participate in this study, and 5260 were either not eligible (n = 4401) or declined to participate (n = 859). When a reason was provided, the majority said that they had no interest in participating in the study (n = 732). Other reasons provided were personal (n = 9), partner/family advice (n = 38), and transportation costs associated with eight contacts (n = 2). A total of 1887 pregnant women consented and completed the baseline survey. Results from three post-intervention surveys (late pregnancy, early postpartum, and late postpartum) and data extracted from health records are included in the analysis. The response rate for the late pregnancy survey was similar by type of ANC: 72.6% for those in Individual ANC and 76.5% for those in Group ANC. Those lost to follow-up at late pregnancy did not differ significantly in baseline characteristics from the rest of the sample. At the early postpartum survey (roughly 2 months postpartum), the response rate fell to its lowest level, with only 66.6% of those in Individual Care returning, compared to 72.6% of those in Group ANC. The response rate rose to its highest level at the late postpartum follow-up, about 6 months after delivery, with 79.4% of Individual Care and 81.2% of Group ANC participants returning. The number of women who returned to take the surveys was somewhat higher than the survey return rates because some women completed surveys outside the allowable cut-off window. Although we allowed latecomers to take the survey, due to recall bias confirmed by our preliminary analysis, their data were not included in the analyses and are noted as intermittent missing. The Consolidated Standards of Reporting Trials (CONSORT) diagram is presented in Fig 1 (S1 Data).

### Participant Characteristics

In this sample, over 40% of participants were pregnant for the first time. At baseline, nearly 60% were between 20–29 years of age, more than 90% were married or in a relationship, and less than 45% of the women did not complete primary school. About 20% of the sample did not have access to any form of electricity or own livestock or land, and nearly 70% experienced food insecurity in the previous 12 months. Just under 38% of pregnant participants were anemic at baseline. None of these individual-level characteristics were significantly different by type of ANC (Table 2).

Our effectiveness results are mapped to the theory of change model [13] in Fig 2. The left section shows the characteristics of the Malawi Group ANC model. The middle captures the processes of structuring and transforming the ANC experience, including our interacting measures of supportive relationships, empowered partnerships, and meaningful services and information. These combine to reinforce a positive ANC experience and desire to return for care. The transformation in care through Group ANC is reflected in wait times, satisfaction with ANC, and more ANC contacts. The right section shows that the ANC experience promotes greater knowledge and actions leading to better outcomes.

### Outcomes

The outcomes are presented in two tables. Table 3 summarizes continuous variables, analyzed using multivariable regression models. It includes point estimates and 99% confidence intervals to indicate the impact of Group ANC, adjusted for covariates. Table 4 presents categorical variables, showing frequencies, percentages, and odds ratios with 99% confidence intervals to compare the likelihood of outcomes between Group ANC and Individual ANC, adjusted for covariates.

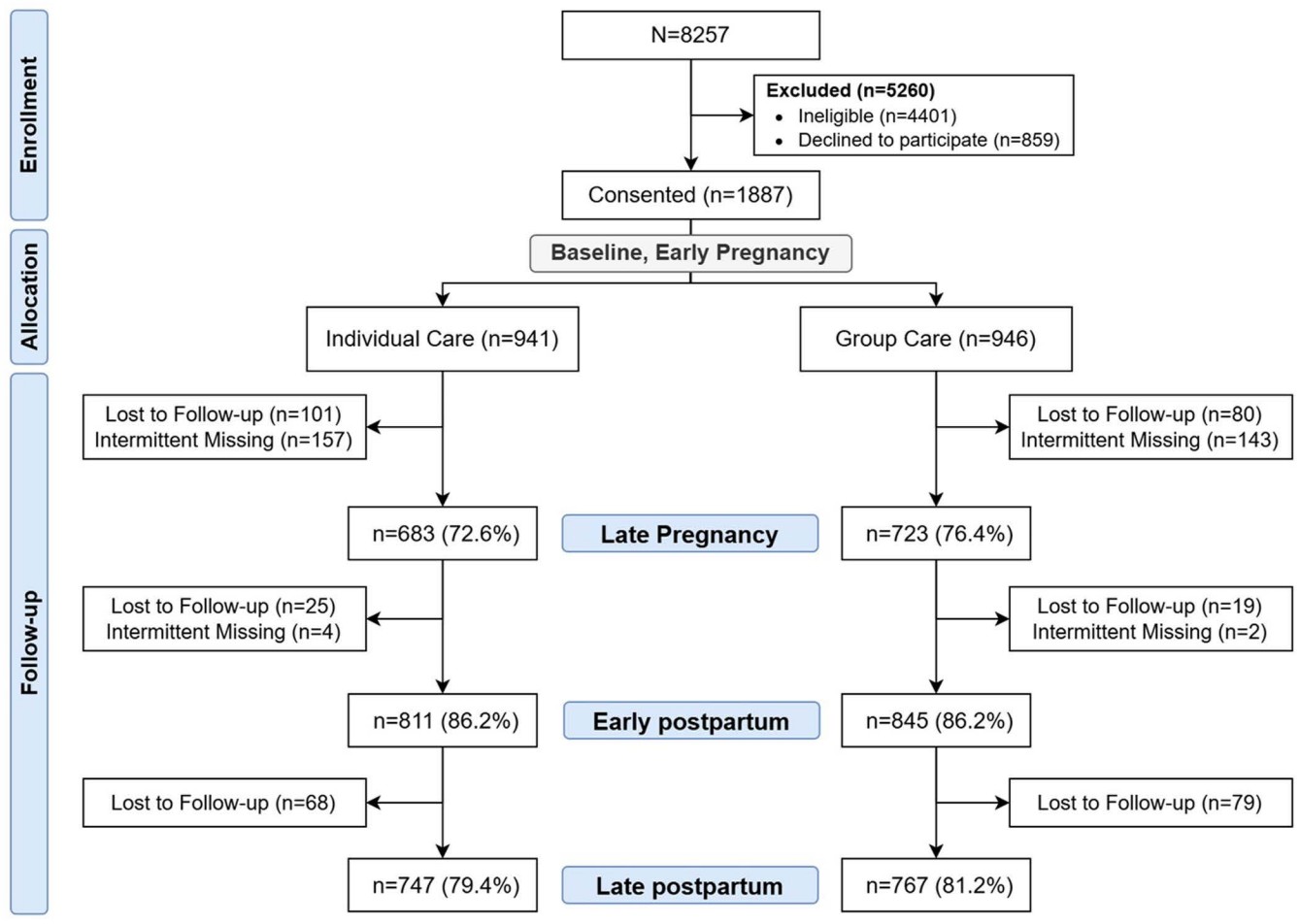

**Fig 1. Consort diagram for the effectiveness trial carried out in Blantyre, Malawi.**

### Self-reinforcing cycle of more and better care

Bivariate and multivariable analyses showed that Group ANC positively related to the four outcomes capturing aspects of the self-reinforcing cycle of more and better care. Our measure of supportive relationships, peer connectedness, was higher among those in Group ANC compared to Individual ANC (Estimate = 1.43; 99% CI = 1.02, 1.84). Women with higher education and who attended clinics serving peri- and urban catchment areas had greater peer connectedness. An indicator of becoming empowered partners in learning and care is captured by the pregnancy-related empowerment scale (PRES). PRES scores were significantly higher for those in Group ANC (Estimate = 2.14, 99% CI = 1.21,3.07), and multiparous women had higher PRES scores. Meaningful services and information were evaluated by women's recall of the number of services and health promotion topics they reported receiving. Women in Group ANC reported receiving more services (Estimate = 0.60, p-value < .0001) and more health promotion topics (Estimate = 0.90, p-value < .0001) than those in Individual ANC. Multiparous women recalled more services and health promotion topics. Group ANC did not relate to having more than one HIV test in late pregnancy, but a lower client-to-midwife ratio was associated with this outcome.

**Table 2. Baseline characteristics by type of ANC.**

| Characteristic, n (%) | Individual ANC (n = 941) | Group ANC (n = 946) | p-value |
|---|---|---|---|
| Primiparous | 395 (42.1) | 379 (40.1) | 0.40 |
| Age category | | | 0.36 |
| 15-19 | 261 (27.7) | 235 (24.8) | |
| 20-29 | 545 (57.9) | 570 (60.3) | |
| 30-35 | 135 (14.3) | 141 (14.9) | |
| In a relationship | 874 (93.1) | 886 (93.9) | 0.49 |
| Education category | | | 0.53 |
| Less than primary | 422 (44.8) | 414 (43.8) | |
| Completed primary | 386 (41.0) | 381 (40.3) | |
| More than primary | 133 (14.1) | 151 (16.0) | |
| Ownership Index | | | 0.82 |
| 0 | 191 (20.3) | 179 (18.9) | |
| 1 | 454 (48.2) | 454 (48.0) | |
| 2 | 254 (27.0) | 271 (28.7) | |
| 3 | 42 (4.5) | 41 (4.3) | |
| Weekly income, quartiles | | | 0.95 |
| Q1 | 213 (24.6) | 214 (24.7) | |
| Q2 | 190 (21.9) | 196 (22.7) | |
| Q3 | 221 (25.5) | 211 (24.4) | |
| Q4 | 243 (28.0) | 244 (28.2) | |
| Food insecure in the last 12 months | 649 (69.0) | 645 (68.3) | 0.74 |

## The ANC experience

Bivariate and multivariable analyses showed a significant positive effect of Group ANC on wait time, satisfaction with care, ANC attendance, and preference for Group ANC in the future. Group ANC participants were 3.26 times more likely to experience a shorter wait time for ANC services. Satisfaction with ANC was higher among those assigned to Group ANC than those in Individual ANC (Estimate = 1.21, 99% CI = 0.07, 2.35). Satisfaction was also associated with more education, older age, higher income, being in a relationship, and greater peer connectedness. Women in Group ANC had a higher mean number of ANC contacts than those in the Individual ANC (Estimate = 0.74, 99% CI = 0.50, 0.98). Women in Group ANC were 1.64 times more likely to complete ANC4 + contacts and 3.27 times more likely to complete ANC8 + contacts. More education and older age were associated with higher attendance. The two clinic-level factors affected the number of ANC contacts and ANC4 +, with rural clinics and lower client-to-midwife ratios associated with increased attendance. Older age was the only additional factor related to attending eight or more visits. Regardless of the assigned type of ANC, most women preferred Group ANC for their next pregnancy (81% in Individual ANC; 95% in Group ANC); this preference was especially marked among women in Group ANC, who were more than 4 times more likely to select Group ANC in the future (OR = 4.40, 99% CI = 2.58, 7.51). A preference for Group ANC related to more peer connectedness and lower client-to-midwife ratios.

## Health knowledge, behaviors, and communication and care

**Knowledge.** Group ANC participants had higher healthy pregnancy knowledge scores than the Individual ANC participants post-intervention (Estimate = 0.42, 99% CI 0.05, 0.79). The 5-item UNAIDS knowledge mean score (Estimate = 0.22, 99% CI 0.10, 0.34) and the dichotomous outcome of whether all items were answered correctly were higher for Group ANC participants than Individual ANC participants. Group ANC participants were 1.93 times more likely

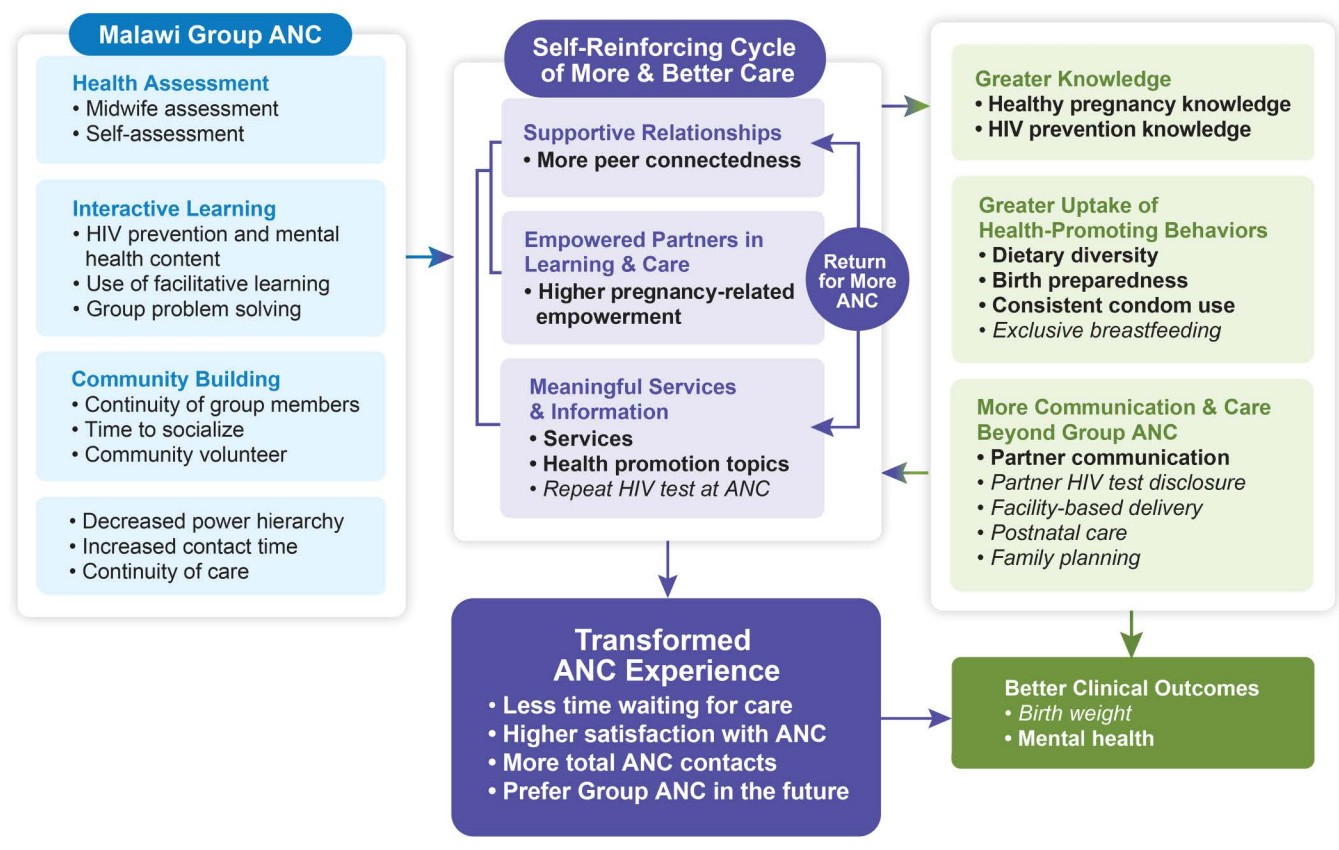

**Fig 2. Malawi Group ANC results mapped to the Grenier et al. theory of change model.** Significant results are bold, and nonsignificant results are italicized.

to answer all five UNAIDS items correctly (99% CI = 1.42, 2.62). For both knowledge measures, baseline knowledge and higher education related to significantly higher late pregnancy knowledge scores.

**Behaviors.** Group ANC positively influenced dietary diversity in late pregnancy, with Group ANC participants having more diverse diets (Estimate = 0.35, 99% CI = 0.17, 0.53). They also reported greater birth preparedness (Estimate 0.32, 99% CI = 0.16, 0.48). Logistic regression indicated that Group ANC participants were 1.65 times more likely to use condoms consistently in the last two months compared to Individual ANC participants (99% CI = 1.06, 2.57). Individual ANC and Group ANC participants did not differ in the proportion who reported exclusive breastfeeding. As expected, the baseline measures (not shown in table) of dietary diversity, birth preparedness, and consistent condom use were significant predictors in regression analyses for their respective outcomes. Greater dietary diversity was also associated with primiparity, owning more items, and attending clinics in peri-urban or urban catchment areas. Being more prepared for birth was related to higher levels of education and owning more items. Consistent condom use was positively associated with less education. Exclusive breastfeeding was related to higher levels of education.

**Communication and care beyond ANC.** Partner communication scores in late pregnancy were higher among Group ANC participants compared to those in Individual ANC (Estimate = 0.21, 99% CI = 0.03, 0.39). Baseline partner communication was the only other significant covariate. Type of ANC did not relate to partner HIV testing and disclosure of results and the only covariate related to this outcome was food insecurity. Individual ANC and Group ANC participants did not differ in the proportion who reported a facility-based delivery, attended postnatal care, or used family planning. The

**Table 3. Type of ANC and regression results for continuous variables adjusted for covariates\* and organized to reflect key Theory of Change processes.**

| Variable, Mean (SD) | Type of ANC | | | Regression | |
|---|---|---|---|---|---|
| | Individual ANC | Group ANC | p-value | Estimated Impact of Group ANC (99% CI) | Significant positive covariates |
| **Self-reinforcing cycle of more and better care** | | | | | |
| *Supportive Relationships* | | | | | |
| Peer connectedness (0-14) | 11.6 (3.70) n=683 | 13.1 (2.04) n=723 | <.0001[1] | 1.43 (1.02, 1.84) | *More peer connectedness was associated with* • More education • Attending a peri-urban or urban clinic |
| *Empowered partners in learning and care* | | | | | |
| Pregnancy-related empowerment (16-64) | 56.48 (7.13) n=683 | 58.65 (6.25) n=724 | <.0001[1] | 2.14 (1.21, 3.07) | *Higher empowerment was associated with* • Being multiparous |
| *Meaningful services and information* | | | | | |
| ANC services (0-9 service) | 7.14 (1.62) n=683 | 7.75 (1.26) n=723 | <.0001[2] | 0.60 (0.39, 0.81) | *More services recalled were associated with* • Being multiparous |
| ANC health promotion topics (0-13) | 11.40 (2.33) n=683 | 12.31 (1.36) n=724 | <.0001[2] | 0.91 (0.65, 1.17) | *More topics recalled were associated with* • Being multiparous |
| **Transformed ANC Experience** | | | | | |
| Satisfaction with ANC (10-50) | 32.8 (7.95) n=683 | 34.9 (8.00) n=723 | <.0001[1] | 1.21 (0.07, 2.35) | *More satisfaction was associated with* • More education • Older age • Being in a relationship • Higher income More peer connectedness |
| Number of ANC contacts | 4.8 (1.91) n=941 | 5.6 (2.00) n=946 | <.0001[1] | 0.74 (0.50, 0.98) | *More contacts were associated with* • More education • Older age • Attending a rural clinic A lower client-to-midwife ratio |
| *Knowledge* | | | | | |
| Healthy Pregnancy Knowledge (0-25) | 16.00 (3.21) | 16.40 (3.24) | <.0123[1] | 0.42 (0.05, 0.79) | *More healthy pregnancy knowledge was associated with* • More education |
| UNAIDS Knowledge (0-5) | 4.20 (0.99) | 4.50 (0.88) | <.0001[2] | 0.22 (0.10, 0.34) | *More UNAIDS knowledge was associated with* • More education |
| *Uptake of Health-Promoting Behaviors* | | | | | |
| Dietary Diversity (0-6) | 3.60 (1.58) | 4.00 (1.53) | <.0001[2] | 0.35 (0.17, 0.53) | *More dietary diversity was associated with* • Being primiparous • More items owned • Attending a peri-urban or urban clinic |
| Birth Preparedness (0-5) | 3.70 (1.37) | 4.10 (1.19) | <.0001[2] | 0.32 (0.16, 0.48) | *More preparedness was associated with* • More education • More items owned |
| *Communication and care beyond ANC* | | | | | |
| Partner Communication (0-6) | 4.40 (1.48) | 4.71 (1.34) | <.0002[2] | 0.21 (0.01, 0.41) | *None of the covariates were significant* |
| **Clinical outcomes** | | | | | |
| Mental health, in late pregnancy | 7.57 (4.89) n=617 | 7.16 (4.91) n=655 | 0.1372[1] | -0.61 (-1.20, -0.02) | *Higher mental distress score was associated with* • Less education |

[1]Student's t-test; [2]Mann-Whitney test

\*Adjusted for all covariates: age, parity, in a relationship, education, food security, ownership index, income quartile, client-to-midwife ratio, and community catchment area

only factor related to attending postnatal care was a lower client-to-midwife ratio. Younger age was positively associated with the uptake of family planning. A total of 17 individuals had a positive pregnancy test in the late postpartum: 9 in Individual ANC and 8 in Group ANC.

## Clinical Outcomes

Based on the literature [17–25,101], two clinical outcomes could have been affected by type of ANC, birth weight and mental health. Type of ANC did not relate to low birth weight (LBW), and logistic regression showed that attending a rural clinic was the only covariate associated with an increased likelihood of LBW. The impact of Group ANC on mental distress in late pregnancy was assessed using the Self Reporting Questionnaire (SRQ) [99] and was not significant in bivariate analysis. However, after controlling for baseline score and the covariates, multivariable analysis showed that women in Group ANC had lower mental distress scores than those in Individual ANC (Estimate = −0.61, 99% CI = −1.20, −0.02). Less education was the only covariate associated with higher mental distress. As expected, several clinical outcomes did not differ by type of ANC, and none approached statistical significance (see Table 4). Less than 2.1% of participants experienced a miscarriage. Hypertension in pregnancy was experienced by less than 6% of participants. Anemia was experienced by about 47% of participants. Cesarean section rates were about 11%, and 98% of women had a singleton birth. There was only one maternal death from initial recruitment through the end of the study, around six months postpartum, that the death occurred in the Group ANC arm of the study. From surveys, health records, and personal follow-up by study personnel, we identified 38 infant deaths, 16 in Individual Care and 22 in Group ANC.

## Effect sizes of outcomes: Group ANC versus individual ANC

We plotted the continuous and categorical results on forest plots to show the effect sizes of Group ANC's impact on these outcomes. Partial eta-squared, the proportion of variance in the outcome variable explained by Group ANC, was used for continuous outcomes [102]. Group ANC had effects of medium size (0.01 < partial eta-squared < 0.06) on 12 outcomes in multivariable regression analyses. The effect sizes of Group ANC for the remaining outcomes were below 0.01 and deemed small. An odds ratio forest plot using 99% CI for ORs was used to illustrate the effect sizes of Group ANC on all categorical outcomes [103]. Among the outcomes for which the Group ANC had a significant impact (i.e., the 99% CI did not cross the vertical line of 1), ANC4+ and UNAIDS knowledge had medium effect sizes, while the effect sizes for a wait time of less than one hour and ANC8 + were large, at greater than 3 (Fig 3).

## Discussion

This study, conducted in Blantyre District, Malawi, investigated the impact of a Centering-based Group ANC model of care implemented with high fidelity. We mapped the outcomes developed for this study to the theory of change model (see Fig 2). Our measures were developed before the publication of the theory of change model and do not fully capture all the nuances of the model. However, our approach highlights substantial differences in outcomes by type of ANC and links them directly to the proposed processes that differentiate women's ANC experiences.

In the theory of change, the core features of Group ANC are supportive relationships, empowered partners in learning and care, and meaningful services and information, all of which work synergistically to develop a self-reinforcing cycle of more and better care and transform the overall ANC experience. Our study had five outcomes reflecting the self-reinforcing cycle, and four were more positive for women in Group ANC: peer connectedness, empowerment, and number of ANC services and health promotion topics. More than three-quarters of participants had more than one HIV test during ANC, but the type of ANC did not relate to this outcome. This rate is substantially higher than previous reports [104], which is a positive trend, as repeat testing of HIV-negative women every 3 months is recommended due to the high risk of seroconversion in pregnant women in areas with high HIV prevalence. The self-reinforcing cycle of higher quality of care and more time devoted to impactful health promotion leads to a transformed ANC experience. Indicators such as reduced wait

**Table 4. Type of ANC and logistic regression results for categorical variables adjusted for covariates\* and organized to reflect key Theory of Change processes.**

| Variable, n (%) | Type of ANC | | | Logistic Regression | |
|---|---|---|---|---|---|
| | **Individual ANC** | **Group ANC** | *p*-value | **Odds Ratio (99% CI)** | **Significant positive covariates** |
| **Self-reinforcing cycle of more and better care** | | | | | |
| *Meaningful services and information* | | | | | |
| Repeat HIV test at ANC[a] | 559 (77.6) n = 720 | 599 (79.2) n = 756 | 0.4566[1] | 1.14 (0.81, 1.60) | *Repeat HIV Test was associated with* • A lower client-to-midwife ratio |
| **Transformed ANC Experience** | | | | | |
| Waited less than one hour for ANC services | 453 (66.3) n = 683 | 625 (86.4) n = 723 | <.0001[2] | 3.26 (2.29, 4.63) | *A shorter wait was associated with* • Being food secure |
| ANC4 + contacts | 750 (79.7) n = 941 | 819 (86.6) n = 946 | <.0001[2] | 1.64 (1.18, 2.27) | *Attending ≥4 contacts was associated with* • More education • Older age • Attending a rural clinic • A lower client-to-midwife ratio |
| ANC8 + contacts | 55 (5.8) n = 941 | 160 (16.9) n = 946 | <.0001[2] | 3.27 (2.15, 4.98) | *Attending ≥8 contacts was associated with* • Older age |
| Prefer Group ANC in a future pregnancy, % yes | 553 (81.0) n = 683 | 690 (95.4) n = 724 | <.0001[2] | 4.40 (2.58, 7.51) | *Group ANC preference was associated with* • More education • More peer connectedness • A lower client-to-midwife ratio |
| *Knowledge* | | | | | |
| UNAIDS knowledge, all correct | 351 (51.4) n = 683 | 475 (65.7) n = 723 | <.0001[2] | 1.93 (1.42, 2.62) | *Answering all questions correctly on UNAIDS knowledge was associated with* • More education |
| *Health-Promoting Behaviors* | | | | | |
| Consistent condom use in late pregnancy[b] | 75 (17.2) n = 437 | 130 (26.4) n = 492 | 0.0007[2] | 1.65 (1.06, 2.57) | *More consistent condom use was associated with* • Less education |
| Exclusive breastfeeding | 454 (62.9) n = 722 | 491 (65.6) n = 749 | 0.2849[2] | 1.11 (0.84, 1.48) | *Exclusive breastfeeding was associated with* • More education |
| *Communication and care beyond ANC* | | | | | |
| Partner HIV status disclosed[c] | 594 (74.3) n = 799 | 644 (75.9) n = 849 | 0.4782[1] | 1.13 (0.82, 1.56) | *Partner HIV status disclosure was associated with* • Experiencing food insecurity |
| Facility-based delivery | 791 (97.8) n = 811 | 816 (96.8) n = 844 | 0.2222[2] | 0.71 (0.31, 1.62) | *Facility-based delivery was associated with* • More education • Attending a peri-urban or urban clinic • A lower client-to-midwife ratio |
| Attended postnatal care | 620 (76.5) n = 810 | 652 (77.2) n = 845 | 0.7662[2] | 1.03 (0.75, 1.41) | *Attending a postnatal care visit was associated with* • A lower client-to-midwife ratio |
| Family planning | 703 (94.1) n = 747 | 726 (94.5) n = 768 | 0.7230[2] | 1.10 (0.62, 1.97) | *Uptake of family planning was associated with* • Younger age |
| **Clinical outcomes** | | | | | |
| Low birth weight[d] | 73 (9.5) n = 766 | 87 (11.0) n = 792 | 0.3443[1] | 1.19[d] (0.77, 1.84) | *Higher risk for low birth weight was associated with* • Attending a rural clinic |
| Miscarriage | 17 (2.1) n = 815 | 10 (1.2) n = 848 | 0.1436[1] | N/A | N/A |
| Hypertension in pregnancy | 48 (5.9) n = 811 | 46 (5.4) n = 845 | 0.6763[2] | N/A | N/A |

*(Continued)*

**Table 4.** (Continued)

| Variable, n (%) | Type of ANC | | | Logistic Regression | |
|---|---|---|---|---|---|
| | Individual ANC | Group ANC | p-value | Odds Ratio (99% CI) | Significant positive covariates |
| Anemia, < 11 g/dl | 318 (46.8) n = 680 | 315 (43.8) n = 719 | 0.2672[2] | N/A | N/A |
| Stillbirth | 8 (1.0) n = 815 | 13 (1.5) n = 848 | 0.3141[2] | N/A | N/A |
| Cesarean Section | 87 (10.7) n = 811 | 95 (11.2) n = 845 | 0.7376[2] | N/A | N/A |
| Singleton birth | 797 (98.3) n = 811 | 825 (98.0) n = 842 | 0.6609[2] | N/A | N/A |
| Maternal death before the end of the study | 0 (0.0) n = 941 | 1 (0.0) n = 945 | 1.0000[3] | N/A | N/A |
| Infant death before the end of the study | 16 (1.9) n = 815 | 22 (2.7) n = 848 | 0.3892[2] | N/A | N/A |

[1]Mann-Whitney test; [2]Chi-squared test; [3]Fisher exact test

*Adjusted for all covariates: age, parity, in a relationship, education, food security, ownership index, income quartile, client-to-midwife ratio, and community catchment area

[a]Women who were living with HIV at baseline were excluded (n = 176); [b] Participants who reported not having sex in the last two months were excluded; [c] Partners who were living with HIV at baseline were excluded (n = 45); [d] The LBW analyses also included women's height, HIV status, tobacco use, anemia < 11g/dl, and dietary diversity as additional covariates reflecting typical risk factors

times, enhanced satisfaction, and increased ANC contacts demonstrate the positive transformation women experienced with Group ANC [1,56,105,106].

The theory posits that the cycle of more and better care also directly links to greater knowledge, more health-promoting behaviors, and more communication and care beyond Group ANC. For post-intervention pregnancy- and HIV-related knowledge, women in Group ANC had higher scores than those in Individual ANC, reflecting the greater effectiveness of interactive learning. Our study also examined four behaviors directly promoted in ANC. Maternal nutrition during pregnancy is a major public health challenge in Malawi and other LMICs [107]. The only source of nutrition education is often ANC, but its impact on nutrition behaviors and outcomes has been limited [108]. This study is the first to report dietary changes related to Group ANC. Birth preparedness and complication readiness is another ANC health promotion topic linked to improved maternal and neonatal outcomes [109,110]. Similar to results from other African-based studies [13,54,111], women in Group ANC in Malawi were better prepared for birth, organizing the necessary supplies and logistics. Women in Group ANC were also more likely to use condoms consistently in late pregnancy, highlighting an increased commitment to HIV prevention. Unlike a trial in Ghana [63], Group ANC did not affect exclusive breastfeeding in Malawi.

We mapped five outcomes to the category of communication and care beyond ANC. Our measure of partner communication was more positive for women in Group ANC than for those in Individual ANC. Like HIV testing at ANC, there was also no effect on partner disclosure; both likely reflect the growing impact of national testing and treatment programs [112]. Facility-based delivery, postnatal care attendance, and uptake of family planning in late postpartum did not differ by type of ANC and may reflect ceiling effects. Malawi has a nearly universal rate of facility-based deliveries [113], and rates of family planning and postnatal care attendance were higher in this study compared to other settings [54,55,114].

Clinical outcomes resulting from transformed experience and knowledge, health-promoting behaviors, and communication and care are the final set of outcomes we present. Notably, Group ANC did not reduce the risk for low birth weight. This finding is consistent with results from Rwanda [56] and Haiti [51], but contrary to studies conducted in the US and Iran, which showed reduced risk for prematurity and LBW [17–25]. This discrepancy may be due to higher rates of first-trimester ANC initiation and reliable ultrasound assessments in the latter studies. Attending Group ANC was associated

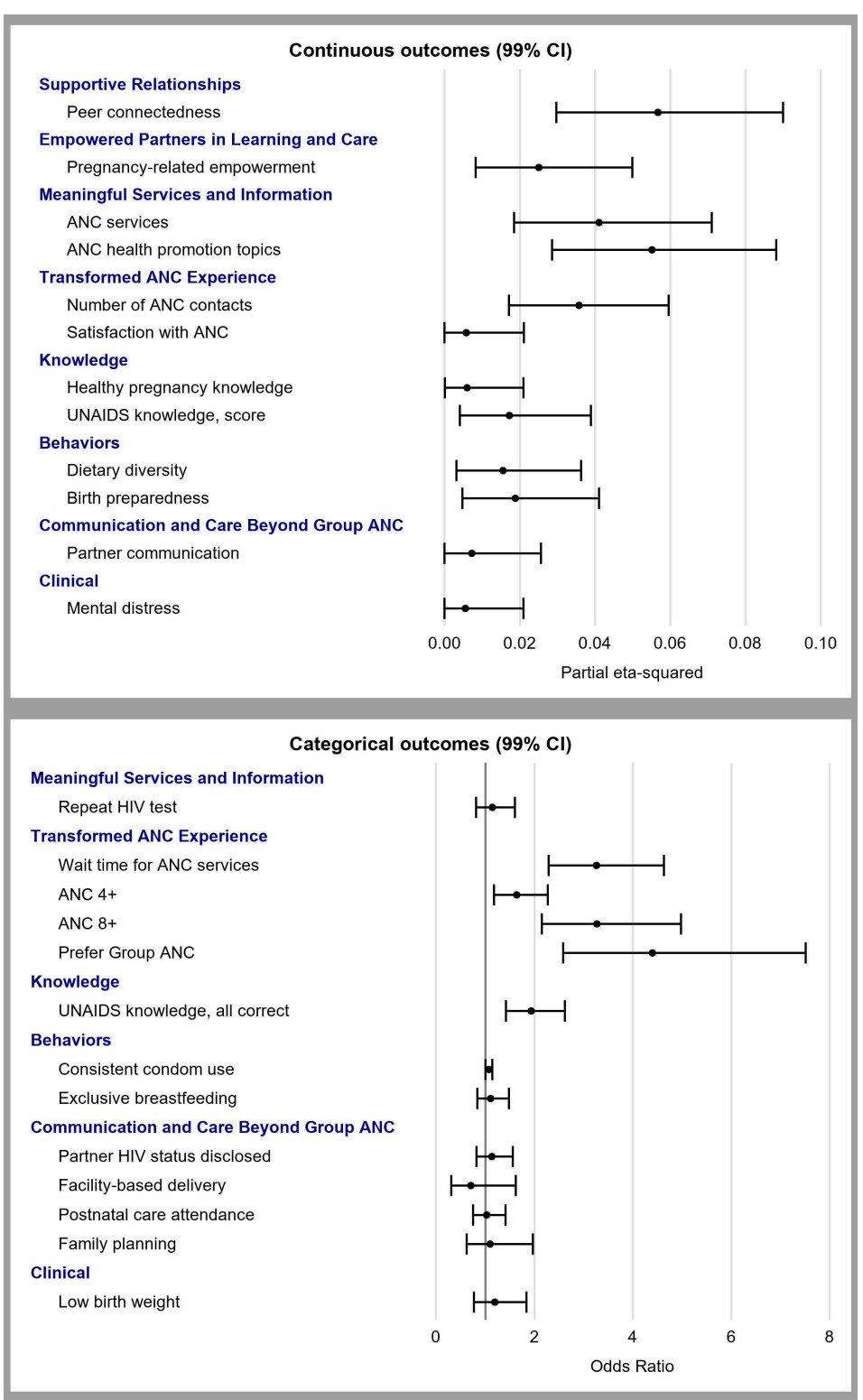

**Fig 3. Forest plots for the effectiveness of continuous (top panel) and categorical (bottom panel) outcomes.**

with reducing mental distress, underscoring its potential contribution to interrupting the cycle of poor maternal mental health and its cascading effects on family well-being [115–118]. To our knowledge, this is the first study of Group ANC in an LMIC to report improvements in mental health.

Several covariates remained significant in multivariable models. Education, a key factor affecting health literacy and behaviors [119–121], was linked to more outcomes than any other covariate. This suggests that training facilitators to be attentive to varying literacy levels can enhance the impact of Group ANC. Age and parity are related to several outcomes and likely reflect accumulated life knowledge and/or previous ANC experience. Our economic indicators of wealth related to three outcomes. Higher income was associated with increased satisfaction with ANC, and item ownership was linked to two health behaviors that require financial resources, birth preparedness and dietary diversity. Catchment area and client-to-midwife ratio significantly influenced various outcomes. As expected, rural catchment areas had a higher proportion of low birth weight infants [122,123]. Peri-urban and urban areas were linked to a higher proportion of births in facilities, greater peer connectedness, and more dietary diversity, reflecting higher quality of care and more diverse food environments in these areas. The client-to-midwife ratio also had an impact on the quality-of-care indicators. A lower ratio was associated with more recall of health promotion topics and more HIV tests during pregnancy, suggesting that midwives with lighter workloads can dedicate more time to these activities. It is also related to more ANC contacts, postpartum visit attendance, and a preference for Group ANC. Adjusted regression models were presented in other cRCTs conducted in Africa [54,56,60,63,111], but detailed information was not provided; therefore, additional analyses are needed to explore the impact of covariates in other large trials. Importantly, the powerful effects of the Group ANC model remain evident even after controlling for covariates. Following recommendations for testing complex interventions [124], this trial incorporated a process evaluation to assess implementation success and features related to sustaining and scale-up in Malawi, and these results are forthcoming.

## Limitations

Despite these promising findings, this study has several limitations. The inaccuracy of gestational age estimates impacted our ability to assess risk for preterm birth. This inaccuracy also affected the ability to align health promotion content with stages of pregnancy, especially for those who gave birth earlier than expected. This study was not designed to track referral pathways of women experiencing morbidity during pregnancy. This limited our ability to explore whether Group ANC led to more appropriate referrals and healthcare use, and fewer emergencies. Further research comparing early problem identification, referrals, and outcomes for women in Group ANC compared to individual ANC would contribute greatly to the literature.

Another limitation is the high probability of spillover effects from two sources. In each clinic, all the midwives were trained and facilitated Group ANC, and these same midwives provided usual care, including Individual ANC. As documented in previous studies, being trained to offer Group care affects provider-client interactions and overall job performance [13,114,125–128]. Women in Individual Care attended more ANC contacts than the national average. However, the specific impact of the spillover from providing the control arm with a schedule cannot be isolated from other ways women in the study may differ from other women who declined to participate. These two spillover effects may have reduced the magnitude of the difference in outcomes observed between Group ANC and Individual ANC.

In addition to the COVID-19 pandemic, this clinical trial was disrupted by several cyclones, including the longest-lasting Category 5 cyclone ever recorded [129,130]. These disasters directly caused morbidities and mortalities as well as closures, travel limitations, supply chain interruptions, displacement due to infrastructural damage from landslides and flooding, and a cholera outbreak. Because ANC is an essential service, mitigation strategies were put in place so that usual ANC and Group ANC could continue to be offered. Analyses of national datasets showed a significant association between disasters and less healthcare utilization [131]; however, we do not account for how these disruptions may have impacted our results [132,133].

Several challenges common to implementing Group ANC have been identified, such as finding clinic space for group sessions, scheduling, willingness to devote two hours to each ANC session by providers and pregnant women, and financial constraints [134]. Committed leaders and providers can usually find solutions to these practical problems. From the earliest days that group healthcare was offered, women's personal preferences for one-on-one care and their concerns about physical and emotional exposure in a group setting have been recognized as potential issues [11]. Well-established strategies to reduce exposure or overhearing of what is said during the one-on-one midwife consultations and the ground rules established by each group to preserve confidentiality of personal disclosures during Group ANC sessions mitigate confidentiality and privacy issues [92–94,114,135]. A more serious potential concern is that, although no evidence suggests higher adverse events in Group ANC, there were slightly more stillbirths and infant deaths, and the only maternal death occurred in Group ANC. However, with a sample size of over 1,500 women, the probability is well below statistical significance, making it unlikely that these differences have clinical significance.

## Conclusion

Since 2016, Group ANC and Group Care Beyond Birth have expanded rapidly into more than 30 LMICs. Increasing evidence, including this study, demonstrates that the Group ANC is equal to or more effective than traditional care. The momentum for reimagining how ANC can more fully address women's healthcare needs is growing. However, given the many challenges of integrating a transformative model of care into complex health systems, scale-up has been limited.

This study of Group ANC in Malawi is the first clinical trial with individual randomization in an LMIC. Results document a wide variety of improved outcomes for women who participated in Group ANC compared to those in Individual ANC, making a substantial contribution to the growing body of evidence that group care is effective in lower resource settings and providing support for the power of interactive learning in a group. The successful integration of two health promotion topics not typically addressed at ANC, primary prevention of HIV and perinatal mental health, demonstrates the flexibility of the group care model to address context-specific priorities and be responsive to emerging individual and community-level needs. This is also the first study to show increased dietary diversity, more partner communication, consistent condom use, and improved mental health. This is the first analysis to use the Grenier et al. theory of change [13] to guide the presentation of the outcomes, highlighting the key processes that differentiate women's experiences with group care and providing evidence of a transformational experience.

This study supports efforts to scale and sustain group care in LMICs [136–139]. With political will, health systems can improve the quality of care to address women's unmet social and health needs. Based on study results, the Malawi Ministry of Health is integrating Group ANC into district-level trainings while exploring nationwide adoption. This momentum positions LMICs to improve healthcare experiences for women, infants, and families in Malawi and globally.

## Supporting information

**S1 Data. CONSORT checklist.**
(PDF)

## Acknowledgments

We are inspired by the midwives, community volunteers, and administrators at participating clinics because they showed dedication to improving reproductive, maternal, newborn, and child health outcomes through the implementation of this evidence-based model of care. We are indebted to the women who agreed to participate and appreciate the time and effort involved in moving science forward. We appreciate the Malawi Ministry of Health for its continued efforts to improve the healthcare system in Malawi. A special thank you to Group Care Global, Carrie Klima, and Amy MacDonald for partnering with us and ensuring that training was top-notch.

## Author contributions

**Conceptualization:** Crystal L. Patil, Kathleen F. Norr, Li C. Liu, Elizabeth Chodzaza, Genesis Chorwe-Sungani, Elizabeth T. Abrams, Sharon S. Rising, Ellen Chirwa.

**Data curation:** Esnath Kapito, Genesis Chorwe-Sungani, Ursula Kafulafula, Dhruvi R. Patel.

**Formal analysis:** Kathleen F. Norr, Esnath Kapito, Li C. Liu, Xiaohan Mei, Allissa Desloge, Dhruvi R. Patel, Heidy Wang, Jocelyn Faydenko, Ellen Chirwa.

**Funding acquisition:** Crystal L. Patil, Kathleen F. Norr, Li C. Liu, Elizabeth Chodzaza, Genesis Chorwe-Sungani, Ellen Chirwa.

**Investigation:** Crystal L. Patil, Ellen Chirwa.

**Methodology:** Crystal L. Patil, Kathleen F. Norr, Li C. Liu, Xiaohan Mei, Ellen Chirwa.

**Project administration:** Crystal L. Patil, Esnath Kapito, Elizabeth Chodzaza, Genesis Chorwe-Sungani, Ursula Kafulafula, Allissa Desloge, Ellen Chirwa.

**Supervision:** Crystal L. Patil, Esnath Kapito, Elizabeth Chodzaza, Genesis Chorwe-Sungani, Ursula Kafulafula, Elizabeth T. Abrams, Ellen Chirwa.

**Validation:** Crystal L. Patil, Esnath Kapito, Li C. Liu, Xiaohan Mei, Allissa Desloge.

**Visualization:** Crystal L. Patil.

**Writing – original draft:** Crystal L. Patil, Kathleen F. Norr, Li C. Liu, Xiaohan Mei, Elizabeth T. Abrams, Ashley Gresh, Ellen Chirwa.

**Writing – review & editing:** Crystal L. Patil, Kathleen F. Norr, Esnath Kapito, Li C. Liu, Xiaohan Mei, Elizabeth Chodzaza, Genesis Chorwe-Sungani, Ursula Kafulafula, Elizabeth T. Abrams, Allissa Desloge, Ashley Gresh, Rohan D. Jeremiah, Dhruvi R. Patel, Anne Batchelder, Heidy Wang, Sharon S. Rising, Ellen Chirwa.

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
