## [Decision Letter · Decision Letter 0]

24 Mar 2025

PONE-D-24-59174Group antenatal care positively transforms the care experience: Results of an effectiveness trial in MalawiPLOS ONE

Dear Dr. Patil,

Thank you for submitting your manuscript to PLOS ONE. After careful consideration, we feel that it has merit but does not fully meet PLOS ONE’s publication criteria as it currently stands. Therefore, we invite you to submit a revised version of the manuscript that addresses the points raised during the review process.

**I want to extend my congratulations to you and the co-authors on a well-written manuscript. In addition to the relevant issues raised by the two reviewers, I have several issues in your methods and results as summarized below: -**

**In the methods, "Statistical Analysis" section from line 248, your regression analysis approaches are not clearly described. The last column of your regression results (from Table 3) indicate that the coefficients are either "Estimates" or odds ratios. However, for the continuous variables, even if the estimates are from the logistic regression models, the relative risks are still odds ratios.****If view of #1 above, did you use two separate regression models ("Regression/Logistic Regression"), and if so, why? Logistic regression would have been sufficient, unless your outcome was continuous. Otherwise, were the variables in the first column your outcomes? please use footnotes (see #3 below) to provide necessary descriptions for clarity. In the "Findings" of your abstract, you reported the parameters as "Estimates". Are these ORs, or both ORs and Coefficients? ORs are also estimates. Please see how to make this clearer for the readers.****Indicate which covariates were adjusted for in your analyses. Consider using table footnotes to provide additional information about your regression analyses.****For the clinical outcomes in Table 6, why not also report the findings as you have done for Tables 3-5? Yet, this would not apply where there are very small cases for regression analyses as the parameters will be inestimable. Nevertheless, Reviewer 2 raises some issues with respect to these results.**

We look forward to receiving your revised manuscript.

Kind regards,

Innocent B. Mboya, Ph.D.

Academic Editor

PLOS ONE

**Journal Requirements:**

1. When submitting your revision, we need you to address these additional requirements. Please ensure that your manuscript meets PLOS ONE's style requirements, including those for file naming. The PLOS ONE style templates can be found at https://journals.plos.org/plosone/s/file?id=wjVg/PLOSOne_formatting_sample_main_body.pdf and https://journals.plos.org/plosone/s/file?id=ba62/PLOSOne_formatting_sample_title_authors_affiliations.pdf 2. Please include a complete copy of PLOS’ questionnaire on inclusivity in global research in your revised manuscript. Our policy for research in this area aims to improve transparency in the reporting of research performed outside of researchers’ own country or community. The policy applies to researchers who have travelled to a different country to conduct research, research with Indigenous populations or their lands, and research on cultural artefacts. The questionnaire can also be requested at the journal’s discretion for any other submissions, even if these conditions are not met. Please find more information on the policy and a link to download a blank copy of the questionnaire here: https://journals.plos.org/plosone/s/best-practices-in-research-reporting. Please upload a completed version of your questionnaire as Supporting Information when you resubmit your manuscript. 3. In the online submission form, you indicated that your data is available only on request from a third party. Please note that your Data Availability Statement is currently missing the name of the third party or institution. Please update your statement with the missing information. 4. When completing the data availability statement of the submission form, you indicated that you will make your data available on acceptance. We strongly recommend all authors decide on a data sharing plan before acceptance, as the process can be lengthy and hold up publication timelines. Please note that, though access restrictions are acceptable now, your entire data will need to be made freely accessible if your manuscript is accepted for publication. This policy applies to all data except where public deposition would breach compliance with the protocol approved by your research ethics board. If you are unable to adhere to our open data policy, please kindly revise your statement to explain your reasoning and we will seek the editor's input on an exemption. Please be assured that, once you have provided your new statement, the assessment of your exemption will not hold up the peer review process. 5. We are unable to open your Figure file (no. 2). Please kindly revise as necessary and re-upload.

Reviewers' comments:

Reviewer's Responses to Questions

**Comments to the Author**

1. Is the manuscript technically sound, and do the data support the conclusions?

Reviewer #1: Yes

Reviewer #2: Yes

2. Has the statistical analysis been performed appropriately and rigorously? 

Reviewer #1: Yes

Reviewer #2: Yes

3. Have the authors made all data underlying the findings in their manuscript fully available?

Reviewer #1: Yes

Reviewer #2: Yes

4. Is the manuscript presented in an intelligible fashion and written in standard English?

Reviewer #1: Yes

Reviewer #2: Yes

5. Review Comments to the Author

**Reviewer #1:**  This paper reports final results from randomized trial of one-on-one antenatal care (the current and most common model for ANC in this setting) versus group ANC. The authors used a centering-based group ANC approach and report on both individual and clinic level outcomes.

MINOR:

Abstract: why 4 OR 8 visits?

Line 66: remove attend the

Line 73: clarify that standard initial ANC visit is one-on-one/individual

Line 75-76: are these times inclusive of wait times?

Line 74-78: is group ANC conducted at the same health centre, or offsite?

Line 80: are there non-pregnant group members in ANC? are guardians or male partners part of group ANC?

Line 140-142: move to discussion

Line 124: most HTC among women in Malawi probably happens through/during ANC. Is HTC not HIV prevention? Or because it happens at a different part of the health center, is it not considered ANC?

Throughout: Consistency... group ANC or Group ANC?

Line 240: I suggest "Malawi Kwacha" equivalent of five USD...

Line 283: suggest allowable (or per protocol) rather than acceptable

Line 193: what is the sample size estimate for >99% power? It is in the protocol paper but would be important to indicate it here as well, so that readers know where the 1887 is in the ballpark or not.

Line 171: what was the eligibility criteria? Or do readers need to visit the protocol paper?

MAJOR:

1. Paper is well-written. It could be reviewed and shortened for readability, by removing repetition. For example, group ANC is described both in the introduction and methods. I think the authors' idea was to describe group ANC in general in the intro and then the Malawi adaptation in the methods?

2. It would be helpful to clarify that no individual level ANC assessments were missed due to participation in group ANC.

3. It would be good to state explicitly (if true) that women were not compensated for ANC visits (individual or group) unless it was a visit during which ACASI survey was planned. This would address a potential source of bias.

4. The 5260 should please be broken down by refused versus ineligible (it's in Fig 1, but please add to text). More information on both should be provided IF available. Why ineligible? And did anyone who refused/declined provide a reason? For example, if someone said I don't have time for group ANC or a friend was in group ANC before and didn't like it, then that might suggest that a certain selection bias that could be indicated in the limitations.

5. Line 220: 1) Was there iterative adaptation of the study interventions or ACASI over the 3 year period? 2) Did 'standard of care' or usual ANC change at all during this period? Did the facilities host other ANC quality improvement activities, such as ANC ultrasound studies/demonstration projects, during this same period, to your knowledge?

6. This is not a costing or cost-effectiveness study but could the authors broadly describe the costs/inputs required for group ANC? Is it only the training of facilitators and co-facilitators? Update of ANC registers to reflect group ANC?

**Reviewer #2: ** Congratulations on a well written paper concerning an important issue, I enjoyed reading it. However I do have some concerns. Table 6 concerns important clinical outcomes and yet there is no demonstration of statistical significance. Both stillbirth and infant death seem to be considerably higher in the group ANC cohort and the only maternal death occurred in this group. Above all, women who embark on pregnancy want their babies to be born alive and survive, and of course to survive intact themselves and yet you have not considered if this difference is of statistical significance nor discussed it. I feel that this must be addressed before the paper should be published.

I would also like to see some consideration as to if and how women in the group cohort were afforded the opportunity to have confidential discussions about matters of concern that they did not wish other women to know about. Also more description as to how women were examined in a manner that preserved their privacy and dignity considering you state that most examinations occurred in the group context. Although you have clearly described the many advantages that you found for the group cohort I would like to have read a more nuanced consideration of possible disadvantages, which may have included challenges to patient confidentiality.

6. PLOS authors have the option to publish the peer review history of their article (what does this mean? ). If published, this will include your full peer review and any attached files.

**Do you want your identity to be public for this peer review?** For information about this choice, including consent withdrawal, please see our Privacy Policy .

Reviewer #1: No

Reviewer #2: No

---

## [Author Response · Author response to Decision Letter 1]

15 May 2025

We thank you and the reviewers for their thoughtful review of our manuscript. The abstract now shows how estimates are expressed. To summarize, in the revised submission, we clarified the use of regression and logistic regression models and made the necessary distinctions between continuous and categorical outcomes. We added footnotes indicating which covariates were included to be more easily accessible to readers. We have added statistical tests for clinical outcomes and provided the p-values. We adjusted the text to distinguish between general and specific descriptions of Group ANC. We added information to address concerns about confidentiality and privacy by expanding on these topics in the methods and discussion sections. We added to the discussion about potential spillover effects, external disruptions, and limitations related to rare clinical events. For readability and conciseness, we tried to reduce repetitive statements without losing key messages. We feel that we have thoroughly addressed critiques and suggestions, including minor edits and rephrasing (e.g., language consistency). We provide detailed responses and locate major changes to each reviewer’s critiques in the table uploaded to the system.

---

## [Editor Report · Decision Letter 1]

21 May 2025

Group antenatal care positively transforms the care experience: Results of an effectiveness trial in Malawi

PONE-D-24-59174R1

Dear Dr. Patil,

We’re pleased to inform you that your manuscript has been judged scientifically suitable for publication and will be formally accepted for publication once it meets all outstanding technical requirements.

Kind regards,

Innocent B. Mboya, Ph.D.

Academic Editor

PLOS ONE
---

## [Editor Report · Acceptance letter]

PONE-D-24-59174R1

PLOS ONE

Dear Dr. Patil,

I'm pleased to inform you that your manuscript has been deemed suitable for publication in PLOS ONE. Congratulations! Your manuscript is now being handed over to our production team.

Kind regards,

on behalf of

Dr. Innocent B. Mboya

Academic Editor

PLOS ONE